# Intellectual Property Pledge Financing and Enterprise Innovation: Based on the Perspective of Signal Incentive

**Weixiu Li [1] and Bo Li [2],***

[1] Longshan Honors School, Shandong University of Finance and Economics, Jinan 250014, China; liweixiu@mail.sdufe.edu.cn
[2] School of Management, Tianjin University of Technology, Tianjin 300384, China
* Correspondence: lb2088@email.tjut.edu.cn

**Abstract:** As a driving force behind urban sustainable development, enterprise innovation has become an increasingly important issue in the digital economy. In this context, a financing model called intellectual property pledge financing (IPPF) has been widely implemented, potentially promoting innovation output in developed countries. However, for countries with relatively low levels of intellectual property (IP) protection, the impact of IPPF on enterprise innovation is divergent, as it may lead to patent signal failure. China's implementation of IPPF since 2008 provides an ideal quasi-natural experiment for researching IPPF in such countries. Using panel data of China's listed companies from 2007 to 2017, we employ the staggered Difference-in-Differences (DID) method to examine the impact of IPPF on enterprise innovation. The results demonstrate a significantly positive impact overall, with a more pronounced effect in urban areas characterized by high intellectual property protection and digitalization. Various robust tests, including event study, Bacon decomposition, and propensity score matching (PSM), were conducted. Additionally, our findings suggest that IPPF facilitates enterprise innovation by expanding external credit resources and optimizing internal management from the perspective of open innovation (OI). It signals banks and investors to provide favorable credit support externally, helps alleviate managerial myopia, and increases manager risk preference internally. These results offer empirical evidence and suggestions for promoting IPPF as a means to stimulate enterprise innovation and achieve urban economic sustainable development.

**Keywords:** intellectual property pledge financing; enterprise innovation; urban sustainable development; digital economy; staggered DID

## 1. Introduction

Urban sustainability has emerged as a global challenge and a focal point of attention, particularly in the era of rapid globalization and digitization. In this context, enterprise innovation plays a crucial role in achieving sustainable urban development. By consistently introducing new ideas, technologies, and business models, enterprise innovation fosters creativity and competitiveness in the urban economy. This, in turn, creates opportunities for the city's prosperity and sustainable growth. However, innovation processes are costly, requiring continuous and substantial investment in research and development (R&D) by enterprises. Moreover, developing countries such as China face significant limitations in terms of their R&D expenditure [1]. Hence, a new financing model is necessary for most enterprises, particularly in developing countries.

The Intellectual Property Pledge Financing Policy (IPPF) has been developed for over a century in Europe and the United States. In 2013, 38% of US enterprises utilized patents as collateral for funding, covering 20% of R&D costs and innovation output [2]. Specifically, IPPF is a financing method that establishes the legal position of patent pledges. Under this policy, IP rights holders pledge their legally owned patents, registered trademarks, copyrights, and other assets to secure funds from financial institutions such as banks. Effective communication of the significance and value of patents is crucial. China implemented

IPPF relatively later, with the official commencement in 1995 through the People's Republic of China's Guarantee Law. In 2007, China had only 682 registered patents for pledges, with a total value of less than CNY 5 billion. However, after 2008, China launched the patent pledge pilot program, conducting five batches of projects to accelerate the pace of IPPF. By 2022, the national patent and trademark pledge financing amount reached CNY 486.88 billion, sustaining a three-year growth rate exceeding 40%. Additionally, mature patent pledges systems, such as the Beijing model, Shanghai model, and Wuhan model, were developed.

The existing literature primarily focuses on the implementation of IPPF in developed countries such as the United States, while there is limited research on IPPF in developing countries. It is important to acknowledge that economic factors, such as GDP, market size, and regulations, differ significantly between developed and developing countries. Consequently, developing countries often lack awareness of IP protection, and they may struggle to take appropriate actions to safeguard enterprises' intellectual property within their borders. As a result, the implementation of IPPF in these countries can have diverse and potentially detrimental impacts.

Unlike other financing policies, IPPF serves a distinct function by promoting incentives for innovation [2]. IPPF can send a positive signal to the external market, thereby providing additional support for enterprise innovation. However, there are arguments suggesting that IPPF may result in patent signal failure, particularly in countries with a low level of IP protection [3]. As the primary lending institutions, banks encounter challenges in both assessing the value of the intellectual property and its potential fluctuations, as well as the risk associated with enterprises' ability to repay loans when they become due [4]. In the patent pledge market, certain enterprises may engage in fraudulent practices by obtaining loans based on "fake patents" that lack genuine technical value. This behavior amplifies the overall risk within the patent pledge financing system and can potentially lead to "patent signal failure". As the instances of free-riders increase, financial institutions become more cautious and hesitant to provide loans, which is detrimental to enterprises seeking funds. Consequently, the impact of IPPF on enterprise innovation remains uncertain in developing countries.

Furthermore, IPPF can send signals not only to the external market but also to the internal operations of enterprises [5]. By transforming patents into valuable assets for enterprises, IPPF can incentivize management decisions and potentially influence the level of managerial myopia and risk preferences regarding long-term innovation investments. However, existing literature primarily focuses on the alleviation of financing constraints for enterprises, neglecting the perspective of open innovation. To address this research gap, we aim to examine the quasi-natural experiment of China's IPPF and investigate its effects and influencing mechanisms on enterprise innovation.

This study may offer several contributions. Firstly, we employ the staggered Difference-in-Differences (DID) method to conduct empirical analysis, enabling us to obtain causal and general insights into the impact of IPPF on enterprise innovation, beyond relying solely on individual case studies. Secondly, we specifically focus on IPPF in developing countries, aiming to provide valuable references for improving IP-related policies in similar contexts. Thirdly, we integrate the Upper Echelon Theory (UET) and Signal Theory (ST) with the perspective of open innovation (OI) to explore the underlying mechanisms, thereby extending the application of these theories and serving as a reference for future research. Lastly, to enhance the credibility of our findings, we employ robust testing methods such as event study and Goodman–Bacon decomposition, considering the inherent estimation errors associated with staggered DID. These rigorous tests have bolstered the reliability and validity of our conclusions.

## 2. Literature Review

### 2.1. Factors That Influencing Enterprise Innovation

Enterprise innovation requires consistent and stable financial support, setting it apart from other investment activities. The fear of disclosure and theft makes enterprises hesitant to share proprietary knowledge and operational information with external investors, leading them to shoulder the high costs of proprietary expenses independently [6]. However, enterprises encounter additional difficulties when relying solely on internal resources for innovation. Chesbrough and Bogers [7] introduced the theory of open innovation to address these issues. Open innovation is an innovation model that involves the utilization of external resources, partner knowledge, and creativity, in conjunction with an organization's internal innovation capabilities, to foster collaborative innovation. Unlike the traditional closed innovation model, open innovation transcends these boundaries by actively engaging external partners, customers, suppliers, academia, and communities in the creation and sharing of knowledge, technology, and resources. This collaborative approach facilitates a more expansive and interconnected innovation ecosystem. In a comprehensive review, Bogers et al. [8] summarize various perspectives and levels of analysis in the field of open innovation research. They propose a four-dimensional model for open innovation, encompassing knowledge introduction, knowledge externalization, partnerships, and open innovation networks. In a study conducted by Laursen et al. [9], the impact of openness on the innovation performance of British manufacturing enterprises was examined. The research findings suggest that open innovation, as opposed to closed innovation, significantly enhances the innovation performance of enterprises. This highlights the importance of collaborating with external partners and facilitating the flow of knowledge. Furthermore, researchers have also directed their attention to the implementation and management aspects of open innovation.

From the viewpoint of open innovation, external resources play a crucial role in fostering enterprise innovation. They enable enterprises to acquire and integrate valuable assets such as knowledge, technology, capital, talent, and market insights. By leveraging a diverse range of external sources, enterprises can enhance their innovation efforts and drive growth. These sources encompass suppliers, partners, customers, university research institutions, and startups. By leveraging these diverse resources, enterprises can fuel their innovation activities and enhance their competitive edge in the market [8]. The influence of external resources on enterprise innovation manifests across various levels. By acquiring and sharing open resources, enterprises can broaden their scope of innovation opportunities, access external inputs for innovation, and consequently foster the development of enterprise innovation. This process enables enterprises to tap into external knowledge and expertise, facilitating the creation of novel ideas, technologies, and solutions. The utilization of external resources ultimately drives the growth and advancement of enterprise innovation [10]. Powell et al. [11] examined the correlation between cooperation and innovation in the biotechnology sector. The study discovered that enterprises actively establish cooperative networks with external partners to foster the sharing of knowledge and resources, thereby facilitating the generation and dissemination of innovation.

The other factor is internal management. Empirical findings have revealed that internal management factors such as leadership style, organizational culture [12], incentive mechanisms, and knowledge management [13] have a significant impact on organizational innovation. Among them, executive decision-making has a significant impact on organizational learning, resource allocation, and innovation activities [14]. Managers are individuals with limited rationality, and their preferences and emotions can cause psychological biases in the uncertain decision-making process, including short-sighted biases caused by focusing on immediate interests [15]. According to the Upper Echelon Theory (UET), collective decision-making is achieved through individual decision-making, which is based on personal characteristics such as individual values and cognitive foundations. Hence, managerial knowledge networks, values, and psychological preferences have a considerable impact on the enterprise's strategic choices [16]. Based on this theory, em-

pirical research on executive characteristics and enterprise creativity has become very popular. Executives' acquired experiences, such as executive education [17], executive tenure [18], educational background [19], international experience [20], and financial background [21], among others, can influence executives' risk preferences, which in turn affect enterprise investment choices and technological innovation. Age [22], gender, and other fundamental characteristics of executives also have significant and even more lasting effects on executives' behavior.

Based on the review of the above literature, we speculate that IPPF may have an impact on the external environment and internal management of the enterprise, thereby promoting enterprise innovation. Hence, Hypothesis 1 was developed as follows:

**Hypothesis 1 (H1).** *Intellectual Property Pledge Financing Policies can promote enterprise innovation.*

### 2.2. The Effects of IPPF on Enterprises

2.2.1. The Impact of IPPF on External Financial Resources

The discussion on the impact of IPPF on enterprises in the existing literature mainly focuses on directly obtaining financing and alleviating financing constraints [23]. However, alleviating financing constraints is not just about the direct inflow of pledged financing funds. IPPF differs from other financing policies in that it also transmits signals, which is the most notable distinction [24]. As the most important source of innovative financing for enterprises, banks play a crucial role in transmitting information through IPPF and pledge is an important factor in determining bank loans [25]. However, credit constraints on enterprises by banks are severe due to the widespread information asymmetry and lack of collateral in enterprises, as well as the lack of incentives for banks to actively collect enterprise information and implement risk management. Kang et al. [26] believe that in patent pledge financing activities, banks hope to obtain information about the number of patents held by enterprises to better evaluate their future technological innovation value and avoid adverse selection. Hochberg et al. [3] also considered that patent contracts require more extensive disclosure of patent information, which can directly transmit more indications about an enterprise's potential development capabilities. According to Hussinger et al. [27], clarifying the value of knowledge patents can reduce the ambiguity of enterprise R&D information, reduce market participants' perceptions of negative R&D results, direct the flow of financial capital into enterprise R&D innovation, and ultimately improve enterprise innovation efficiency and market value. The experience of developed countries shows that IPPF broadens enterprises' financing channels and increases external financing. Based on the above analysis, Hypothesis 2 is developed:

**Hypothesis 2 (H2).** *Intellectual Property Pledge Financing Policies promote enterprise innovation through expanding bank loans.*

As a financing method, IPPF can provide signals not only to banks but also to investors, influencing their emotions and investment decisions [28]. Especially for investors with a long-term vision, the signaling effect of intellectual property pledge financing will be more significant [29]. Early research mainly focused on the relationship between intellectual property rights and corporate stock prices. Barth et al. [30] conducted a study on Australian listed companies from 1991 to 1995 and discovered a significant correlation between the revaluation of intangible assets and stock prices. They believed that disclosing intellectual property asset information to investors would affect their expectations of the enterprise's future growth and profitability, and stock prices would be a direct reaction of investors to the enterprise's evaluation. Moreover, Hall et al. [31] found that the number of patent applications is closely related to the market value of enterprises. Hence, if the market recognizes the current status, methods, and degree of information disclosure of enterprises, it should give a positive response to the voluntary disclosure of enterprises. Disclosing information about intellectual property assets is a positive signal that an enterprise sends to the market,

allowing investors to understand the company's innovation ability, market competition, and future development potential, and make informed evaluations and decisions.

**Hypothesis 3 (H3).** *Intellectual Property Pledge Financing Policies promote enterprise innovation by enhancing investor sentiment.*

2.2.2. The Impact of IPPF on Internal Management

In modern enterprises, it is common to have a governance structure with a separation of powers. When management personnel formulate and execute strategic goals and operational decisions, their personal characteristics and risk preferences become apparent, which influences the decision-making of the enterprise. From the perspective of enterprise internal management, research on IPPF and this issue is often ignored, especially manager behavior. Managerial myopia can generally be divided into two categories: self-interest short-sightedness and catering short-sightedness. Self-interest myopia arises from executives' need to satisfy their interests, including compensation [32], position, and reputation [33]. External market pressures, mainly including investor sentiment, short-term investor preferences [34], analyst tracking [35], etc., trigger executive myopia. The main way to alleviate executive myopia is through external governance; furthermore, improving the external environment can effectively dispel the external drive of managerial myopia.

IPPF can serve as a long-term incentive mechanism that aligns the personal interests of executives with the long-term development of the enterprise [36]. Executives are inclined to adopt long-term strategies and decisions to enhance enterprise performance and ensure the successful recovery of pledged patents. It can provide additional sources of funding for the enterprise, particularly for those facing capital constraints. These extra funds can be used to support research and development, innovation, and long-term investments, encouraging executives to prioritize the enterprise's long-term growth. It helps alleviate the short-termism pressure faced by executives. Through patent pledging, executives can focus more on innovation and long-term value creation, reducing the impact of short-term fluctuations on decision-making [37].

However, the path of influence of IPPF based on enterprise management has been neglected. Through the incentive effect of converting patents into funds, IPPF affects the management behavior of listed companies, making managers realize the importance of R&D innovation and thus affecting the enterprise innovation investment. Based on the above analysis, Hypothesis 4 is developed:

**Hypothesis 4 (H4).** *Intellectual Property Pledge Financing Policies promote enterprise innovation through the path of alleviating managerial myopia.*

Enterprise innovation involves high risks, and failure to innovate often results in significant sunk costs and consequences, especially when the enterprise has insufficient cash flow. However, there is a significant conflict between risk-taking and the career development and interests of managers. Managers are responsible not only for the enterprise but also for formulating strategies that align with its development. The more risks they take, the greater their worries about their career and financial stability. Therefore, managers who are willing to bear risks are more likely to promote innovative activities [38].

As an incentive mechanism, IPPF encourages executives to take higher risks in innovation and value creation. Executives may be inclined to invest in research and development projects with higher risk and potential returns, aiming to enhance the company's competitiveness and long-term growth prospects [39]. Patent pledging is often closely associated with a company's innovation activities and intellectual property. These areas typically involve higher uncertainty and risks but can also offer higher rewards. Therefore, executives may be more inclined to make investments and decisions in these high-risk, high-reward domains. By aligning executives' interests with the enterprise's performance and share-

holder interests, when executives gain additional benefits through patent pledging, they may be more willing to take on higher risks in pursuit of greater returns.

IPPF offers objective environmental support that can increase the tolerance of enterprise innovation, thereby enhancing the risk tolerance of managers. IPPF will also provide positive signals to the management of enterprises, improve the level of internal informatization, and ultimately lead to an increase in manager risk aversion [40].

**Hypothesis 5 (H5).** *Intellectual Property Pledge Financing Policies promote enterprise innovation through the path of improving manager risk preference.*

## 3. Data, Variables, and Methodology

The methodology is presented as follows. First, we estimate the model with baseline regression using the staggered DID method (see Section 4.1). To overcome the potential bias of the staggered DID, we perform various robustness tests, including event study, Goodman–Bacon decomposition, propensity scores matching (PSM), replacing variables, and placebo tests (see Section 4.2). Then we perform heterogeneity analysis by categorizing the samples based on urban patent protection and digital development degree (see Section 4.3) and explain why IPPF could affect enterprise innovation from the perspective of open innovation (see Section 4.4). Our framework and steps are presented in Figure 1.

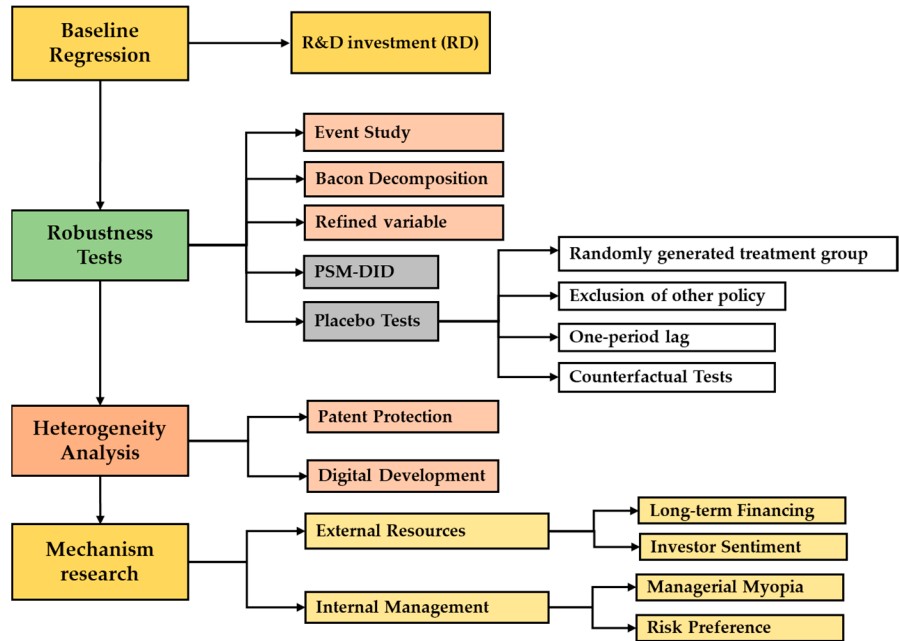

**Figure 1.** Research framework.

### 3.1. Data Source

We chose the listed companies in China as research samples, and the research interval is 2007–2017. The data were mainly gathered from the China Stock Market and Accounting Research Database (CSMAR). We removed the delisted companies and enterprises that are in the financial sector and have been listed for less than one year. Meanwhile, we removed ST, *ST, and SST enterprises and enterprises with abnormal data and serious missing data. Among them, ST represents the enterprise's two consecutive years of operating losses. *ST represents the enterprise that has suffered losses for three consecutive years and is subject to delisting warnings. SST represents the enterprise that has been operating at a loss for two consecutive years and has not yet completed the share reform. Finally, we winsorized all of the continuous variables at a 1% level.

### 3.2. Variables Description

#### 3.2.1. Enterprise Innovation

The dependent variable is enterprise innovation (RD). Existing research generally measures the innovation input or output level as proxy variables of enterprise innovation. Since innovation output is more influenced by exogenous factors and not only influenced by enterprise management, we primarily used innovation input as the proxy variable of enterprise innovation. Following the usual practice [41], we measured the enterprise's innovation input by dividing the R&D expenditure by total assets. In Section 4.2.4, we will also incorporate the number of granted patents as an alternative measurement for innovation input to test the robustness.

#### 3.2.2. Intellectual Property Pledge Financing

The independent variable in our study is Intellectual Property Pledge Financing (IPPF), which is represented as a binary variable. Specifically, if a city $c$ pilots the policy in year $t$, IPPF is assigned a value of 1 and is assigned a value of 0 in other cases. The policy was piloted in five batches. Table 1 provides the detail of the pilot city in China.

**Table 1.** China's Intellectual Property Pledge Financing Pilot.

| Time | Year | City |
|------|------|------|
| 2008.12 | 2009 | Beijing, Changchun, Nanchang, Xiangtan, Foshan, and Ningxia |
| 2009.09 | 2010 | Chengdu, Wuxi, Wenzhou, Yichang, Guangzhou, and Dongguan |
| 2010.07 | 2011 | Shanghai, Tianjin, Zhenjiang, and Wuhan |
| 2012.10 | 2013 | Bengbu and Weifang |

Note: After August 2016, the state has approved regions for patent pledge financing pilot, but no unified list has been formed. Therefore, the first four batches of pilot regions are considered in this paper. Considering the time lag of the policy, the pilot time for the second half of the year (after June) is defined as the following year.

#### 3.2.3. Control Variables

To mitigate potential bias resulting from missing variables, a set of control variables has been chosen. Enterprise maturity influences investment in enterprise innovation, so we measure with enterprise age (Age) and incorporate it into the model. Enterprise shareholder structure has a significant impact on enterprises' innovation strategy, reflecting the degree of management control over the enterprise. Therefore, we control the shareholder shareholding ratio (TOP1). Independent directors can influence enterprise innovation decisions due to their expertise and cognitive level; hence, we control the percentage of independent directors (IND). Furthermore, we include the return on assets (ROA) as a control variable in the model to account for the fact that enterprises with higher profitability levels are capable of allocating more resources toward innovation expenditures. Additionally, we consider the impact of regional economic development on enterprise innovation by including the logarithmically transformed gross regional product (GDP) as another control variable.

#### 3.2.4. Mechanism Variables

(1) Managerial Myopia. Managerial myopia is difficult to measure due to its subjective nature. In this case, previous research measures it indirectly, that is, mainly from perspectives of investor performance. For instance, Bushee [34] used investor turnover rate or institutional investor holdings, and Lundstrum et al. [42] used the short-term portfolio investment ratio as proxy variables of executive myopia. In addition, direct methods such as questionnaire ratings were used [43]. However, direct methods may have low questionnaire response rates and subjective cognitive bias [44]. The early researchers understand human traits and measure managerial myopia as follows. Using a lexical approach in text

analysis, Brochet et al. [35] calculated the frequency of words related to the "time horizon" stated by managers in U.S. surplus conference calls. However, the Chinese language is extensive and profound, and the multiple expressions of semantics are more complex. Therefore, it is inappropriate to use the above method. Based on the method taken by Brochet et al. [35], we referred to Pogach et al. [45] to employ machine learning techniques for text analysis. Using the Chinese "short-term horizon" word set derived from the aforementioned method, we constructed the manager's short-sighted index (Myopia) using a dictionary-based approach.

(2) Manager Risk Preference. Risk preference concerns the psychological inclination of decision-makers towards uncertain risks in strategic decision-making. It manifests in management's willingness to invest in unknown projects while making business decisions. In enterprise investment decision-making, higher-risk options include trading financial assets, available-for-sale financial assets, and investment real estate [46]. To gauge management's risk preference, we measure the ratio of the annual total amount invested in these three high-risk options to the current year's total assets. A higher ratio suggests a more aggressive risk preference on the part of management.

(3) Bank Financing. Innovation investments of enterprises mainly come from long-term investment, so we use long-term loans (Long) and long-term loans ratio (Longratio) to measure the level of external financing of enterprises.

Tables 2 and 3 present the definitions and descriptive statistics of the main variables, respectively. The samples have been further classified based on the characteristics of enterprises. The corresponding results are presented in Table 4.

(4) Investor sentiment. Investor sentiment is a challenging indicator to measure. The commonly used method to measure it is the survey and questionnaire approach, which involves gathering investor sentiment data through surveys and questionnaires. However, there are evident issues of sample bias associated with this method. Currently, the method of Rhodes-Kropf [47] is widely recognized as an effective approach in this regard. The measurement method is as follows. The market valuation (Tobin q) of an enterprise is separated into the intrinsic value part containing its growth and the market mispricing part. Rhodes-Kropf et al. [47] believe that the enterprise size (Size), leverage ratio (Lev), and profitability ability (ROA) are the most important factors involved in fitting its intrinsic value. In consideration of industry differences and market fluctuations, the following cross-sectional regression was carried out for all companies in each industry of all samples in each year. Referring to Rhodes-Kropf et al. [47], we standardized the residuals to obtain investor sentiment indicators. The calculation equation is shown in Equation (1).

$$Q_{i,t} = \delta_0 + \delta_1 Size_{i,t} + \delta_2 Lev_{i,t} + \delta_3 ROA_{i,t} + \varepsilon_{i,t} \qquad (1)$$

The primary variable definitions in this paper are shown in Table 2.

**Table 2.** Primary variables and explanations.

| Variable Type | Variable Name | Symbol | Measurement |
|---|---|---|---|
| Dependent variable | Enterprise Innovation | RD | R&D investment/Asset. See Section 3.2.1 |
| Independent variable | Intellectual Property Pledge Financing | IPPF | Dummy variable. See Section 3.2.2 |
| Control variables | Economy | GDP | Gross regional product |
| | Profit | ROA | Net profit/total assets |
| | Equity Control | TOP1 | the shareholding proportion of the controlling shareholder |
| | Enterprise Maturity | Lnage | Years of establishment |
| | Independence of Directors | IND | Independent directors/directors |
| | Enterprise Size | Size | Total employees |

**Table 2.** *Cont.*

| Variable Type | Variable Name | Symbol | Measurement |
|---|---|---|---|
| | Managerial Myopia | Myopia | See Section 3.2.4 |
| | Management Risk Prefer | Risk | See Section 3.2.4 |
| Mechanism variables | Bank Financing | Long<br>Longratio | Long-Term Loans<br>Long-Term Loans/Loans |
| | Investor sentiment | IS | See Section 3.2.4 |

Note: We take logarithms and winsorize for absolute variables (GDP, TOP1, Lnage, Size, and Long).

### 3.3. Descriptive Statistics

The descriptive statistics of the main variables in our study are presented in Table 3. We found that our research sample is relatively large, which can, to some extent, ensure the credibility of the results. Moreover, each variable has no large outlier, and the mean is close to the median. Table 4 shows the grouping of samples in the heterogeneity section, and the results show that the two groups are relatively balanced.

**Table 3.** Descriptive statistics.

| Variable Type | Variables | Observations | Mean | Standard Deviation | Min. | Median | Max. | Skewness | Kurtosis |
|---|---|---|---|---|---|---|---|---|---|
| Dependent variable | RD | 16,396 | 0.020 | 0.018 | 0 | 0.017 | 0.094 | 1.648 | 6.686 |
| Independent variable | IPPF | 16,396 | 0.223 | 0.416 | 0 | 0 | 1 | 1.334 | 2.779 |
| Control variables | ROA | 16,396 | 0.038 | 0.059 | 0 | 0.0360 | 0.208 | −1.066 | 8.348 |
| | IND | 16,396 | 0.199 | 0.038 | 0 | 0.200 | 0.500 | 0.553 | 4.603 |
| | TOP1 | 16,396 | 0.352 | 0.153 | 0.003 | 0.332 | 0.900 | 0.506 | 2.769 |
| | Lnage | 16,396 | 2.026 | 0.905 | 0 | 2.303 | 3.332 | −0.849 | 2.732 |
| | Size | 16,396 | 7.573 | 1.430 | 1.386 | 7.552 | 13.220 | −0.0440 | 4.349 |
| | GDP | 16,396 | 0.549 | 0.396 | 0.026 | 0.456 | 1.457 | 0.579 | 2.170 |
| Mechanism variables | Myopia | 25,337 | 0.103 | 0.094 | 0 | 0.082 | 1.553 | 2.051 | 12.37 |
| | Risk | 25,964 | 0.0340 | 0.0800 | 0 | 0.00300 | 0.981 | 15.38 | 350.6 |
| | Long | 21,101 | 0.152 | 0.857 | 0 | 0.005 | 32.950 | 2.081 | 7.252 |
| | Longratio | 21,101 | 0.056 | 0.087 | 0 | 0.015 | 0.417 | 4.444 | 29.26 |
| | IS | 22,590 | 0 | 1.023 | −5.310 | −0.164 | 6.630 | 1.919 | 9.869 |

**Table 4.** Enterprise characteristics.

| Background Information | Characteristics | Frequency | Percentage |
|---|---|---|---|
| IP Protection | High | 15,933 | 61.09 |
| | Low | 10,150 | 38.91 |
| Digital Development | High | 15,094 | 57.87 |
| | Low | 10,989 | 42.13 |

Note: We group by means. These variables will be mentioned in the heterogeneity analysis section.

### 3.4. Econometric Model

Since the pilot cities started IPPF at different times, we use the staggered DID method with two-way fixed effects of panel data to verify the impact of IPPF on enterprise innovation. The empirical model is designed as the Equation (2).

$$RD_{i,t,j,p} = \beta_0 + \beta_1 IPPF_{i,t} + \sum \beta\, Control + \gamma_i + \mu_t + v_j + \gamma_p + \varepsilon_{i,t,j,p} \tag{2}$$

where *RD* is the innovation input of enterprises; *IPPF* is the policy variable; *Control* is a series of control variables; subscripts *i,t,j* and *p* denote the enterprise, year, industry, and province, respectively; $\gamma_i$, $\mu_t$, $v_j$, $\gamma_p$ represent the fixed effects of the individual (enterprise),

time (year), industry, and province, respectively; $\varepsilon_{i,t,j,p}$ is the random disturbance term; $\beta_0$ is the constant term. The core coefficient in this model is $\beta_1$, which captures the impact of IPPF on enterprise innovation.

## 4. Results and Discussions

### 4.1. Baseline Regression

The estimated results of Equation (2) are presented in Table 5. In column (1), the estimated coefficient, without adding any control variables, is 0.002, signifying a positive association at a 5% significance level. Then, we gradually add control variables to the model. In column (6), the estimated coefficient for IPPF on enterprise innovation is 0.0017, consistently positive and relatively stable compared to the estimated coefficients in columns (1) to (5). Thus, it can be concluded that IPPF significantly enhances enterprise innovation, supporting the acceptance of H1.

**Table 5.** Regression Results of Intellectual property pledge financing policy on Enterprises Innovation.

| Variables | RD | | | | | |
|---|---|---|---|---|---|---|
| | (1) | (2) | (3) | (4) | (5) | (6) |
| IPPF | 0.0020 ** | 0.0020 ** | 0.0020 ** | 0.0019 ** | 0.0016 ** | 0.0017 ** |
| | (2.71) | (2.71) | (2.72) | (2.62) | (2.14) | (2.19) |
| ROA | | −0.0004 | −0.0004 | −0.0004 | −0.0004 | −0.0004 |
| | | (−1.27) | (−1.27) | (−1.27) | (−1.29) | (−1.29) |
| IND | | | −0.0001 | −0.0002 | 0.0018 | 0.0019 |
| | | | (−0.02) | (−0.03) | (0.32) | (0.33) |
| TOP1 | | | 0.0026 | 0.0019 | 0.0004 | 0.0004 |
| | | | (0.71) | (0.52) | (0.11) | (0.12) |
| Lnage | | | | −0.0008 | −0.0029 *** | −0.0029 *** |
| | | | | (−1.58) | (−4.37) | (−4.35) |
| Size | | | | | 0.0004 | 0.0004 |
| | | | | | (0.90) | (0.94) |
| GDP | | | | | | −0.0005 |
| | | | | | | (−0.19) |
| Year fixed effects | Yes | Yes | Yes | Yes | Yes | Yes |
| Enterprise fixed effects | Yes | Yes | Yes | Yes | Yes | Yes |
| Province fixed effects | Yes | Yes | Yes | Yes | Yes | Yes |
| Industry fixed effects | Yes | Yes | Yes | Yes | Yes | Yes |
| Constant | 0.0196 *** | 0.0197 *** | 0.0188 *** | 0.0206 *** | 0.0220 *** | 0.0222 *** |
| | (101.22) | (101.24) | (9.92) | (9.23) | (6.52) | (5.52) |
| Observations | 15,035 | 15,035 | 15,035 | 15,035 | 13,801 | 13,782 |
| Within R-squared | 0.7620 | 0.7620 | 0.7621 | 0.7622 | 0.7681 | 0.7680 |
| F Statistics | 7.3653 | 4.3918 | 2.3488 | 2.8317 | 9.6157 | 9.3523 |

Note: t-values are reported in parentheses. The robust standard errors are clustered at the province level. *** and ** represent significance at the levels of 1% and 5%, respectively (The table below will not be reiterated).

### 4.2. Robustness Tests

#### 4.2.1. Event Study

The fundamental assumption of the DID model requires that enterprise innovation between treated and controlled groups will develop with the relatively same trend before the shock of IPPF. However, many researchers have noticed that an important potential issue with staggered DID is the heterogeneous treatment effects, where the same treatment has varying effects on different individuals. This difference may manifest in two dimensions, the length of time after receiving treatment or the group receiving treatment, at different time points. Therefore, the event study method is used to observe the changing trend of enterprise innovation estimation coefficient before and after the implementation of IPPF. Referring to Beck et al. [48], we first use the traditional two-way fixed effects (TWFE) to estimate. Then, according to Borusyak et al. [49], we incorporate two types of heterogeneous robust estimators into the model, and the results are presented in Figure 2. The figure illustrates that the estimated coefficients for both the treatment group and the control group before the implementation of IPPF were not statistically significant, indicating that the parallel trend test is successful and validates the results of the baseline regression.

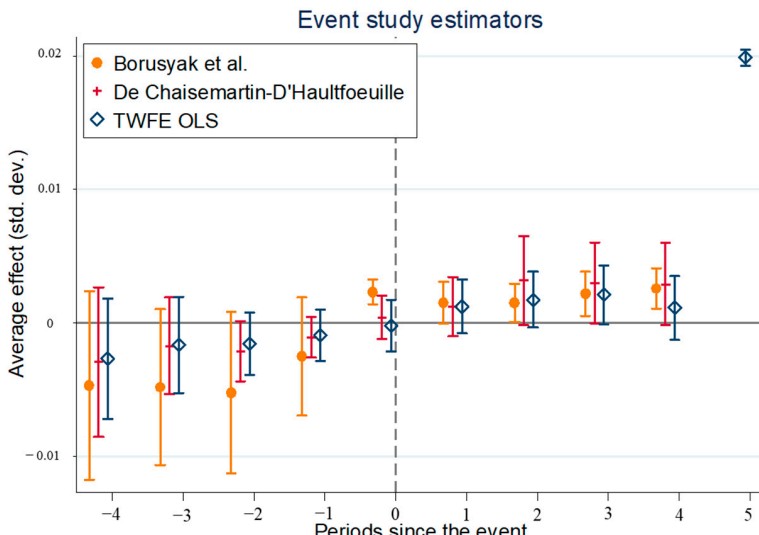

**Figure 2.** Result of the event study. The vertical axis represents the difference in innovation between the treatment and control groups before the policy (i.e., IPPF) implementation. The horizontal axis represents the years before and after the IPPF. The solid lines are the 95% confidence interval of the estimated coefficients in different methods.

### 4.2.2. Bacon Decomposition

We conducted a robustness test using Goodman–Bacon decomposition to verify the accuracy of our estimation results [50], as shown in Figure 3. The panel data set was split into three groups based on processing time: the earlier group treatment, the later group treatment, and the never-treated group. And the red line represents the policy effectiveness of the overall sample. Our analysis indicates that the majority of policy effects originate from the untreated group's counterfactual scores, which served as the control group (represented by triangles in the figure). This implies that the heterogeneous treatment effect did not significantly affect the regression results, thus verifying their robustness and reliability.

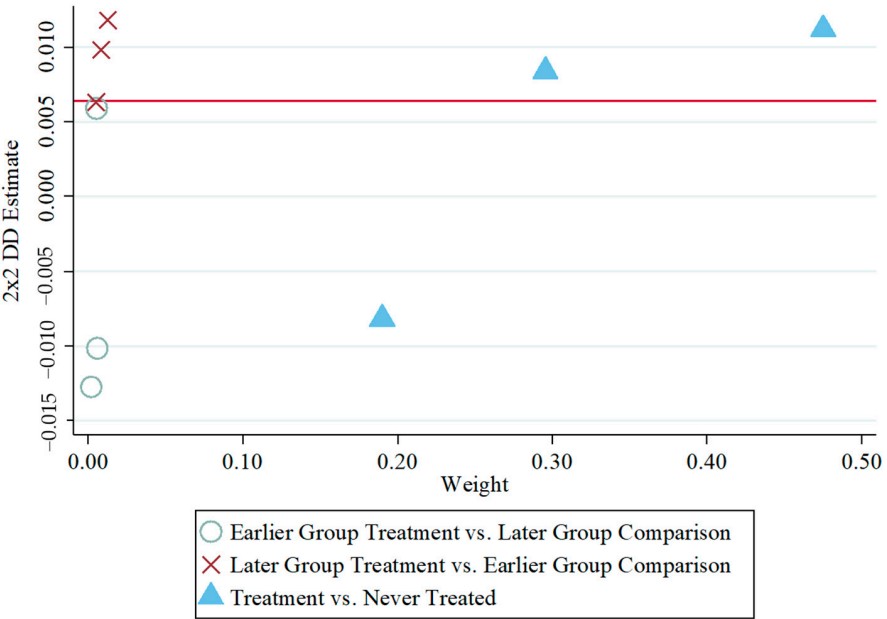

**Figure 3.** Result of Goodman–Bacon decomposition.

### 4.2.3. Propensity Score Matching

Although the DID method addresses the issue of endogeneity to some extent by satisfying the parallel trend assumption, the problem of self-selection may persist. To address this concern, we employ propensity score matching (PSM) by selecting covariates and re-estimating the analysis using the matched sample within the DID framework.

Figure 4a displays the standardized bias across covariates before and after matching, while Figure 4b presents the propensity scores of the treated and control groups. The vertical line represents a deviation rate of 0 in Figure 4a. And the closer to this line, the smaller the deviation rate of the variable. Following the PSM procedure, the results indicate that the standardized bias across covariates is adjusted to within 10%, demonstrating the accuracy of the matching process.

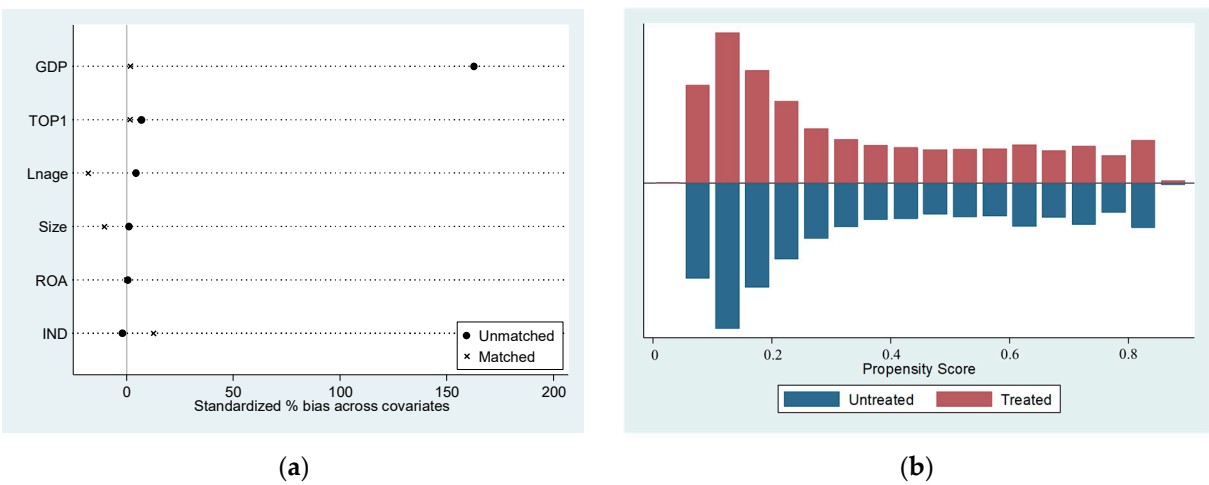

(**a**)             (**b**)

**Figure 4.** Results of the Propensity Score Matching (PSM). (**a**) Standardized bias across covariates; (**b**) common value range density of treatment and control groups.

The regression results are reported in Table 6. After 1:3 nearest neighbor PSM, IPPF still enhances enterprises' innovation with a significance level of 10% and a coefficient of 0.0020. After sequentially adding the same variables as the baseline regression, the conclusion still holds.

**Table 6.** Regression results of the PSM-DID estimation.

| Variables | RD | | | | | |
|---|---|---|---|---|---|---|
| | **(1)** | **(2)** | **(3)** | **(4)** | **(5)** | **(6)** |
| IPPF | 0.0020 * | 0.0020 * | 0.0020 * | 0.0020 * | 0.0020 * | 0.0020 * |
| | (1.94) | (1.94) | (1.95) | (1.88) | (1.88) | (1.90) |
| ROA | | −0.0003 | −0.0003 | −0.0003 | −0.0003 | −0.0003 |
| | | (−1.38) | (−1.43) | (−1.41) | (−1.44) | (−1.44) |
| IND | | | 0.0043 | 0.0043 | 0.0046 | 0.0046 |
| | | | (0.35) | (0.35) | (0.39) | (0.38) |
| TOP1 | | | 0.0023 | −0.0003 | −0.0002 | −0.0002 |
| | | | (0.49) | (−0.05) | (−0.05) | (−0.04) |
| Lnage | | | | −0.0031 *** | −0.0031 *** | −0.0031 *** |
| | | | | (−2.98) | (−2.99) | (−3.08) |
| Size | | | | | 0.0003 | 0.0003 |
| | | | | | (0.47) | (0.48) |
| GDP | | | | | | −0.0008 |
| | | | | | | (−0.12) |
| Year fixed effect | Yes | Yes | Yes | Yes | Yes | Yes |
| Enterprise fixed effect | Yes | Yes | Yes | Yes | Yes | Yes |
| Province fixed effect | Yes | Yes | Yes | Yes | Yes | Yes |
| Industry fixed effect | Yes | Yes | Yes | Yes | Yes | Yes |

**Table 6.** *Cont.*

| Variables | RD | | | | | |
| --- | --- | --- | --- | --- | --- | --- |
| | (1) | (2) | (3) | (4) | (5) | (6) |
| Constant | 0.0206 *** | 0.0206 *** | 0.0190 *** | 0.0260 *** | 0.0240 *** | 0.0245 *** |
| | (58.44) | (58.56) | (8.11) | (7.29) | (5.35) | (3.88) |
| Observations | 6122 | 6122 | 6122 | 6122 | 6122 | 6122 |
| Within R-squared | 0.7801 | 0.7801 | 0.7801 | 0.7811 | 0.7811 | 0.7811 |
| F Statistics | 3.7748 | 2.3999 | 1.2901 | 9.0081 | 7.8062 | 6.7567 |

Note: * represents significance at the level of 10% and *** represents significance at the level of 1%. The PSM-DID method further reduces the bias caused by the self-selection problem.

### 4.2.4. Replacing Variables

To maintain the robustness of results, we measured enterprise innovation from the perspective of output. Following the usual practice, we measured the enterprise innovation in terms of the number of enterprise patents granted (Pat). The regression results are presented in column (1) of Table 7, revealing that the IPPF continues to positively influence the innovation output of enterprises. The estimated coefficient of 15.805 is statistically significant at the 5% level.

### 4.2.5. Placebo Tests

(1) Randomly generated treatment group. To further verify that the improvement of enterprise innovation is caused by the IPPF, rather than other unobservable factors, we refer to Wang et al. [51] to carry out the placebo test. The specific approach involves selecting cities without IPPF implementation as the treatment group for regression analysis within the study period. This random sampling process is repeated 500 times, resulting in the kernel density plot of the estimated coefficient distribution, as depicted in Figure 5. The coefficient of the baseline regression, marked with dotted lines in the figure, is observed to be 0.0017. Notably, a clear distinction is observed between the kernel density of the placebo test and the baseline regression results, enhancing the robustness of the baseline regression analysis.

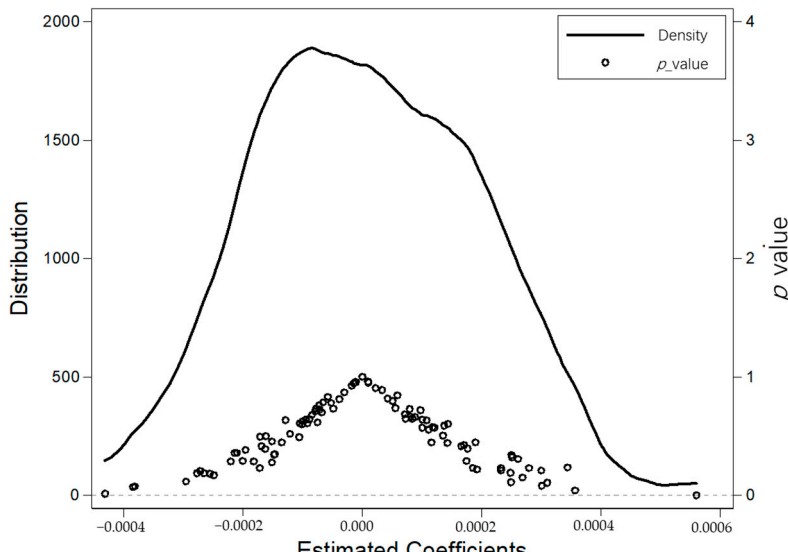

**Figure 5.** Result of the placebo test. The vertical axis represents the density of estimated coefficients of the corresponding pseudo-policy. The black circle represents the distribution of the pseudo-policy coefficients, while the dotted line represents the coefficient of the baseline regression.

(2) Exclusion of other policy effects. Other policies in our research period may have impacts on enterprise innovation as well. For instance, the goal of the Low-Carbon City

Policy (LCC) is to implement a low-carbon economy in the city and create a resource-saving and environmentally friendly society. The LCC was conducted in 2010, and there were 13 pilot cities at first. In 2012, 28 new pilot cities were added. The method of maintaining low energy consumption, pollution, and emissions is to upgrade enterprise technology, which is related to enterprise innovation investment. To eliminate the influence of LCC for the regression, we control the LCC for the robustness test.

The National Intellectual Property Demonstration City Policy (NIPDC) has been piloted to provide a favorable environment and value intellectual property rights in society. In 2012, the first batch of NIPDC were selected as pilot cities. After that, five demonstration cities were selected in 2013, 2015, 2016, 2018, and 2019. The NIPDC may drive the development of urban innovation and further influence enterprise innovation. Hence, we controlled the NIPDC for the regression. Upon excluding the aforementioned policy variable, the regression results remained statistically significant at the 5% level. In both columns (2) and (3) of Table 7, the coefficient estimate for both variables is 0.002.

(3) One-period lag. To address possible endogeneity issues as much as possible, we lag the independent variable (LPolicy) as well as the control variables by one period and then re-estimate the fixed effects on enterprise innovation. The regression results are presented in column (4) of Table 7, indicating that the impact of IPPF on enterprise innovation remains statistically significant at the 5% level, with a coefficient of 0.001.

(4) Counterfactual Tests. If the policy effect remains significant after the year of policy implementation is advanced in the treatment group sample, it indicates that the enhancement effect of IPPF on enterprise innovation is likely to come from other factors. In this part, we assume that the policy is implemented three years earlier in all cases and generate a policy dummy variable (DPolicy). The regression results are presented in column (5) of Table 7, revealing that the coefficients of the core independent variables are not statistically significant. This outcome lends support to the credibility of our findings to some extent.

**Table 7.** Regression results of the other robust tests.

| Variables | (1) | (2) | (3) | (4) | (5) |
|---|---|---|---|---|---|
| | Pat | RD | RD | RD | RD |
| IPPF | 15.805 ** | 0.002 ** | 0.002 ** | | |
| | (2.29) | (2.18) | (2.26) | | |
| LCC | | 0.001 ** | | | |
| | | (2.36) | | | |
| NIPDC | | | 0.0002 | | |
| | | | (0.51) | | |
| LPolicy | | | | 0.001 ** | |
| | | | | (2.21) | |
| DPolicy | | | | | 0.001 |
| | | | | | (1.00) |
| Control | Yes | Yes | Yes | Yes | Yes |
| Year fixed effect | Yes | Yes | Yes | Yes | Yes |
| Enterprise fixed effect | Yes | Yes | Yes | Yes | Yes |
| Industry fixed effect | Yes | Yes | Yes | Yes | Yes |
| Constant | 31.770 *** | 0.020 *** | 0.020 *** | 0.023 *** | 0.018 *** |
| | (3.58) | (9.73) | (9.86) | (5.91) | (7.67) |
| Observations | 17,283 | 15,016 | 15,016 | 12,309 | 15,016 |
| Within R-squared | 0.302 | 0.762 | 0.762 | 0.787 | 0.762 |
| F Statistics | 2.136 | 3.142 | 2.816 | 14.954 | 1.507 |

Note: Pat: Number of enterprise patents; LCC: Low-Carbon City Policy; NIPDC: National Intellectual Property Demonstration City Policy; LPolicy: IPPF lagging for one period; DPolicy: Counterfeiting the policy variable. ** represents significance at the level of 5% and *** represents significance at the level of 1%.

### 4.3. Heterogeneity Analysis

Innovation may bring externalities for enterprises. Specifically, enterprises may find it challenging to stop rivals from copying their creative products. Hence, enterprises suffer larger innovation risks when regional IP rights are not well protected, which may reduce their motivation to innovate. Stronger investigation and prosecution of patent infringement cases indicates better IP protection. Thus, we used the number of urban intellectual property trial settlements to measure the degree of regional IP protection. Then, we divided the enterprises into two groups, namely high IPP (intellectual property protection) and low IPP, based on the mean value of trial settlements. The regression results are displayed in Table 8. In column (1), the regression analysis reveals a substantial positive impact of IPPF on enterprises with stronger intellectual property protection, as indicated by a coefficient of 0.026 and a significance level of 1%. However, the impact on enterprises with low intellectual property protection is not significant, as shown in column (2) of Table 8. Intellectual property protection can reduce instances of patent infringement, including theft and imitation, increase the exclusivity of enterprise patents, create a supportive institutional setting for IPPF, and promote enterprise innovation.

Another factor that influences IPPF is the degree of regional digital development (DD). In regions with high levels of digital development, there are often more developed innovation ecosystems and technology enterprise clusters, making it easier for financial institutions and investors to understand the value of patents and more likely to provide related financing services. Therefore, we grouped the samples using the digital economy level for group regression. Column (3) of Table 8 shows that IPPF does not significantly influence enterprises in low digital development cities. However, with a coefficient of 0.0028 and a significance level of 5%, the regression findings demonstrate that IPPF has a significant favorable influence on enterprises in high digital development cities, as reported in column (4) of Table 8.

**Table 8.** Regression results of the heterogeneity analysis.

| Variables | RD | | | |
|---|---|---|---|---|
| | (1) | (2) | (3) | (4) |
| | High IPP | Low IPP | Low DD | High DD |
| IPPF | 0.026 *** | −0.002 | 0.0003 | 0.0028 ** |
| | (3.62) | (−0.11) | (0.21) | (2.15) |
| Control | Yes | Yes | Yes | Yes |
| Year fixed effect | Yes | Yes | Yes | Yes |
| Enterprise fixed effect | Yes | Yes | Yes | Yes |
| Industry fixed effect | Yes | Yes | Yes | Yes |
| Province fixed effect | Yes | Yes | Yes | Yes |
| Constant | 0.270 *** | 0.076 * | 0.0229 *** | 0.0176 *** |
| | (5.73) | (1.85) | (44.05) | (88.69) |
| Observations | 8271 | 5111 | 6676 | 7834 |
| Within R-squared | 0.798 | 0.749 | 0.7811 | 0.7812 |
| F Statistics | 7.837 | 5.192 | 0.0422 | 4.6392 |

Note: IPP: intellectual property protection; DD: digital development. ***, ** and * represent significance at the levels of 1%, 5% and 10%, respectively.

### 4.4. Influencing Mechanisms

As discussed in Section 2, we have categorized the channels through which IPPF impacts enterprise innovation into internal and external aspects. Figure 6 illustrates our primary analytical approach.

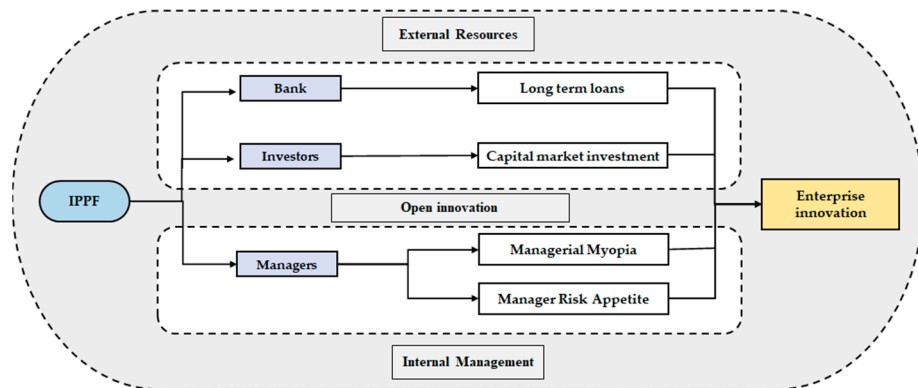

**Figure 6.** The Mechanism of IPPF on Enterprise Innovation.

### 4.4.1. External Resources

IPPF offers a broader understanding of an enterprise's potential and development capacity through the requirement of detailed disclosure of patent information in the process of signing patent contracts [3]. IPPF conveys to banks the information that the enterprise possesses valuable intellectual property assets that can be used as collateral. This signal indicates that the enterprise has a stable innovation capability and valuable intellectual property, which enhances its creditworthiness and loanable amount. When evaluating loan applications, banks consider patent pledging as an important form of collateral. Therefore, IPPF increases the chances of the enterprise obtaining long-term bank loans and may lead to more favorable loan conditions, such as lower interest rates and longer repayment periods. Such policy signals help elevate the company's position and trustworthiness in bank financing, thereby influencing the outcome of its long-term bank loans.

IPPF demonstrates the enterprise's emphasis on its innovation capability and the value of intellectual property, which can have a positive impact on investors' sentiment. This policy sends a signal to potential investors that the company possesses innovative capabilities and potential growth opportunities. This can generate interest and confidence among investors, motivating them to be more willing to invest in the company. Therefore, IPPF may contribute to improving the external financing environment of the enterprise, attracting more investor attention and funding.

To test the external signal mechanism, the following econometric models (3)–(5) are designed.

$$Long_{i,t,j,p} = \gamma_0 + \gamma_1 IPPF_{i,t} + \sum \gamma\, Control + \gamma_i + \mu_t + v_j + \gamma_p + \varepsilon_{i,t,j,p} \tag{3}$$

$$Long\_ratio_{i,t,j,p} = \delta_0 + \delta_1 IPPF_{i,t} + \sum \delta\, Control + \gamma_i + \mu_t + v_j + \gamma_p + \varepsilon_{i,t,j,p} \tag{4}$$

$$IS_{i,t,j,p} = \delta_0 + \delta_1 IPPF_{i,t} + \sum \delta\, Control + \gamma_i + \mu_t + v_j + \gamma_p + \varepsilon_{i,t,j,p} \tag{5}$$

The assessment of bank loans is conducted using two indicators: Long-term loans (Long) and the percentage of long-term loans (Longratio). The regression results for these indicators are presented in columns (1) and (2) of Table 9. Our findings reveal that IPPF significantly increases both the long-term loans and long-term loans ratio of enterprises. These results suggest an improvement in their debt capacity and facilitation of long-term investment. Furthermore, we examined the impact of IPPF on IS using the same methodology as described earlier. The results are presented in column (3) of Table 9, where the estimated coefficients show a significant positive effect at the 5% level. This indicates that IPPF can effectively transmit signals to investors in the capital market, boosting their confidence and providing financial and environmental support for enterprise innovation and development, which verified the previous hypotheses H2 and H3.

There is a positive correlation between investor sentiment and enterprise innovation investment. When investor sentiment is low, the company's financial constraints increase, and the investment scale is reduced. Conversely, when emotions are positive, the investment levels increase [39]. Moreover, the inflow of these investments and loans can increase the source of funds for enterprises, providing more financial support for their research and innovation activities.

### 4.4.2. Internal Management

As mentioned in Section 2.2, IPPF has the potential to influence the internal management behavior of enterprises. IPPF serves as an incentive mechanism, allowing enterprises to utilize their innovative outputs to secure financing. It facilitates the conversion of innovation into economic returns, thereby raising executives' awareness of the significance of innovation for long-term competitiveness and sustainable development. The financial support obtained through IPPF helps mitigate managerial myopia by enabling executives to prioritize long-term strategic planning and investment in innovation, rather than solely focusing on short-term gains. Additionally, IPPF mitigates risks and uncertainties for the enterprise by converting intellectual property into funds. This empowers executives to confidently undertake a certain level of risk, facilitating the pursuit of more ambitious and promising innovative projects.

To verify the above analysis with empirical data, we develop models (6) and (7) to examine the estimated mediating effect of managerial myopia and risk preference.

$$Myopia_{i,t,j,p} = \alpha_0 + \alpha_1 IPPF_{i,t} + \sum \alpha\, Control + \gamma_i + \mu_t + v_j + \gamma_p + \varepsilon_{i,t,j,p} \tag{6}$$

$$Risk_{i,t,j,p} = \delta_0 + \delta_1 IPPF_{i,t} + \sum \delta\, Control + \gamma_i + \mu_t + v_j + \gamma_p + \varepsilon_{i,t,j,p} \tag{7}$$

where *Myopia* is managerial myopia, and the calculating method has been described in Section 3.2.4; the notation of other variables is the same as the Equation (2). The regression results for model (6) are presented in column (4) of Table 9. The estimated coefficient is significantly negative at the 1% level, suggesting that IPPF mitigates managerial myopia. Similarly, the regression outcomes for model (7) are reported in column (5) of Table 9. The estimated coefficient is significantly positive, at the 5% level, implying that IPPF enhances the risk preference of top managers. The above empirical results validate the previous hypotheses H4 and H5.

Innovation is a complex and unpredictable process. As per the Upper Echelons Theory (UET), executive behavior plays a crucial role in shaping the strategies and development of an enterprise. When the management team demonstrates risk aversion or myopic behavior, the strategic decisions tend to be conservative, primarily focused on minimizing uncertainty. This approach leads to limited investment in innovation. On the other hand, when the degree of managerial myopia is low or executives possess a strong risk tolerance, they are more inclined to make proactive decisions. They advocate for a higher tolerance towards R&D innovation and actively promote investment in innovative endeavors. Therefore, IPPF enhances the innovation capability of enterprises by influencing internal management and addressing two aspects: mitigating managerial myopia and enhancing executives' risk preferences.

**Table 9.** Regression results of the external signal and internal signal mechanism.

| Variables | (1) | (2) | (3) | (4) | (5) |
|---|---|---|---|---|---|
| | Long | Longratio | IS | Myopia | Risk |
| IPPF | 0.119 *** | 0.007 ** | 0.083 ** | −0.014 *** | 0.007 ** |
| | (2.81) | (2.18) | (2.07) | (−3.35) | (2.22) |
| Control | Yes | Yes | Yes | Yes | Yes |
| Year fixed effect | Yes | Yes | Yes | Yes | Yes |
| Enterprise fixed effect | Yes | Yes | Yes | Yes | Yes |
| Province fixed effect | Yes | Yes | Yes | Yes | Yes |
| Industry fixed effect | Yes | Yes | Yes | Yes | Yes |
| Constant | 0.021 | −0.514 *** | −0.905 *** | 0.105 *** | 0.038 *** |
| | (1.22) | (−12.13) | (−21.70) | (38.16) | (10.13) |
| Observations | 21,101 | 21,100 | 22,590 | 25,337 | 22,562 |
| Within R-squared | 0.026 | 0.064 | 0.063 | 0.024 | 0.068 |
| F Statistics | 10.021 | 21.503 | 60.601 | 53.921 | 44.776 |

Note: ** represents significance at the level of 5% and *** represents significance at the level of 1%.

## 5. Discussion and Conclusions

### 5.1. Research Conclusions

The primary objective of this study is to examine sustainable solutions for urban development. Specifically, we focus on exploring the relationship between Intellectual Property Pledge Financing (IPPF) and enterprise innovation in China, along with its transmission mechanism. To achieve this, we employed the staggered Difference-in-Differences (DID) model and analyze panel data from Chinese A-share listed companies spanning from 2007 to 2017. The key findings of our empirical analysis can be summarized as follows:

(1) Despite the limitations of China's current intellectual property protection framework and the relatively short implementation period of IPPF, it significantly stimulates innovation among listed firms in China. Our baseline regression results support hypothesis H1, demonstrating that IPPF increases firms' investment in innovation. This finding is robust and supported by additional tests, including event studies, Goodman–Bacon decomposition, PSM-DID, and replacement variables tests.

(2) The impact of IPPF on enterprise innovation exhibits heterogeneity. Specifically, in cities with high intellectual property production and advanced digital development, the effect of IPPF on innovation is more pronounced.

(3) Our mechanism analysis reveals that IPPF promotes enterprise innovation through two channels: enhancing access to external financing resources and optimizing internal management practices. These findings validate hypotheses H2–H5.

(4) Further analysis indicates that IPPF contributes to the enhancement of urban innovation and green innovation, thus promoting sustainable development in cities.

### 5.2. Marginal Contributions and Limitations

This study may make several contributions to the existing research.

(1) Contribution to the literature on IPPF impact: This study adds to the growing body of literature by examining the impact of IPPF implementation in China. Given the late establishment of IPPF in China and its weak intellectual property protection, our research provides valuable insights into the unique context of China and contributes to the assessment of IPPF in the research system. Furthermore, we expand the understanding of IPPF beyond its role in alleviating financial constraints, shedding light on its impact on enterprise internal management. Drawing on the perspective of open innovation theory, our study identifies two key mechanisms of IPPF: external resource allocation and internal management optimization. This offers a fresh research perspective for further exploration of IPPF-related studies.

(2) Adoption of advanced measurement methods for improved robustness: To address the inherent estimation bias in the staggered Difference-in-Differences (DID) approach, we employed heterogeneous robust estimators to conduct event studies, Goodman–Bacon decomposition, and other robustness tests. These findings provide valuable insights for future research on the topic of staggered DID methodology, enhancing the reliability and validity of our results.

(3) Supplementary contribution to the theoretical framework of urban sustainable development: The patent pledge financing policy introduces an innovative financial model that supplements the theoretical framework of urban sustainable development. It emphasizes the significance of intellectual property rights in driving innovation and economic growth, offering a new perspective and research approach to the theory of sustainable urban development. This policy serves as an innovative financial tool that supports and contributes to the sustainable development of cities.

However, this study has several limitations that warrant attention in future research. Firstly, it primarily focuses on the relationship between IPPF and enterprise innovation input, using innovation intensity as a measure. To provide a more comprehensive understanding of enterprise innovation, future studies should develop comprehensive indicators that capture both the input and output dimensions of innovation. Secondly, the measurement of managerial myopia in this study relies on word-frequency information from publicly traded companies, which represents an improvement over previous approaches but may still have limitations. Future research should explore more appropriate measures to accurately capture managerial myopia. Thirdly, while this study acknowledges that the relationship between IPPF and enterprise innovation may involve additional factors, these factors have not been fully considered. Future research should aim to explore and incorporate these potential factors for a more comprehensive understanding of the relationship between IPPF and enterprise innovation. Addressing these limitations will contribute to a more nuanced and comprehensive understanding of the dynamics between IPPF, enterprise innovation, and related factors.

*5.3. Suggestions*

The research presented in this paper provides academic support for understanding the impact of IPPF on enterprise innovation and suggests relevant policy implications. The empirical findings have significant implications for policy-making.

Firstly, it is recommended to gradually expand the scope of the IPPF pilot program and increase financing options for patent pledges. Our study demonstrates a significant positive relationship between enterprise innovation and the IPPF pilot, which not only facilitates enterprise financing but also reduces managerial myopia and enhances employee education. These findings can inform the national promotion of the patent pledge finance model.

Secondly, fostering an innovative culture and improving internal management practices are crucial. Establishing a corporate culture that encourages innovation, motivates employees to propose innovative ideas, and provides necessary support and resources is essential. Additionally, optimizing internal management mechanisms to ensure effective management and implementation of innovative projects, including standardizing innovation processes and implementing incentive and reward mechanisms for innovation teams, is recommended.

Finally, it is essential to ensure the proper implementation of supporting actions for IPPF. Our study reveals a strong correlation between IPPF and the level of regional intellectual property protection. IPPF has a more significant impact on enterprises that benefit from robust local court patent protection. Therefore, creating an environment that emphasizes intellectual property protection and takes action against infringement of enterprise innovation rights is crucial for fostering a conducive environment for enterprise innovation.

These policy recommendations aim to leverage the positive impact of IPPF on enterprise innovation and create an enabling ecosystem that supports sustainable and effective innovation in enterprises.

**Author Contributions:** Conceptualization, W.L. and B.L.; methodology, B.L.; software, W.L.; validation, W.L. and B.L.; formal analysis, W.L.; investigation, W.L.; resources, B.L.; data curation, W.L.; writing—original draft preparation, W.L.; writing—review and editing, W.L. and B.L.; visualization, W.L.; supervision, B.L.; project administration, B.L.; funding acquisition, B.L. All authors have read and agreed to the published version of the manuscript.

**Funding:** The research project is supported by the Natural Science Foundation of Shandong Province, China (ZR2019MG040), and the Ministry of Education of Humanities and Social Science project, China (19YJAZH063).

**Institutional Review Board Statement:** Not applicable.

**Informed Consent Statement:** Not applicable.

**Data Availability Statement:** The official website of the CSMAR database is https://www.gtarsc.com (accessed on 14 June 2023).

**Conflicts of Interest:** The authors declare no conflict of interest.

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
