# Peer review of "Intellectual Property Pledge Financing and Enterprise Innovation: Based on the Perspective of Signal Incentive"

_sustainability, doi:10.3390/su151310448_

Round 1
Reviewer 1 Report
Review Report
Intellectual Property Pledge Financing and Enterprise Innovation: Based on the Perspective of Signal Incentive
1. The paper examines a significant issue/mechanisms that offer a contribution to the literature of sustainable financing. It is indeed a major step forward that China is concerned about “intellectual property pledge financing.”
2. The paper is well organised and the literature review is relevant to the topic of the paper.
3. The methodology is very well implemented in terms of using the staggered DID and robustness tests.
4. In addition, the Research process and framework fit both the objective and the data being examined in the paper.
5. In p. 285, I do not think that figure 2 is presenting the facts that well. I would recommend replacing it with a table that explains the numbers sufficiently.
6. Again, it is much better, and clearer that Figure 3 is to be replaced by a table. It is strongly recommended that the table includes the measurement of each control variable.
7. Table 2 can replace Figure 3 sufficiently, given that a column is to be added to include the measurement of each control variable.
8. In table 3, it is quite standard to report further indicators such as standard error, Skewness and Kurtosis.
9. The econometric model (2) is very well presented. Nevertheless, I am afraid that standard tests should have been performed before reaching the final estimation regression equation. These tests include linearity Vs non-linearity (RESET test), Fixed Vs Random Effects (using Hausman test), Heteroskedasticity tests (using Breusch-Pagan/ Cook-Weisberg test).
10. Although, tables 7 and 8 show that the authors have reached the estimations assuming fixed effects, there must be evidence that fixed rather than random effect is the appropriate estimation that fits the data.
11. The tables do not show the significance (p-value) of each model. For example, in table 7, model 5 is insignificant, thus no further discussion is required. The same is true in table 8, model 4. And in Table 10 model 2. In these cases, the authors must discuss (preferably with proper references) the implications of those insignificant models.

Author Response
Responses to the reviewers
Intellectual Property Pledge Financing and Enterprise Innovation: Based on the Perspective of Signal Incentive
Reviewer 1:
|
Comments |
Responses |
||||||||
|
1-4. The paper examines a significant issue/mechanisms that offer a contribution to the literature of sustainable financing. It is indeed a major step forward that China is concerned about “intellectual property pledge financing.”The paper is well organised and the literature review is relevant to the topic of the paper. The methodology is very well implemented in terms of using the staggered DID and robustness tests. In addition, the Research process and framework fit both the objective and the data being examined in the paper. |
Thank you for your encouragement , reviewers! |
||||||||
|
5-7. In p. 285, I do not think that figure 2 is presenting the facts that well. I would recommend replacing it with a table that explains the numbers sufficiently. Again, it is much better, and clearer that Figure 3 is to be replaced by a table. It is strongly recommended that the table includes the measurement of each control variable. Table 2 can replace Figure 3 sufficiently, given that a column is to be added to include the measurement of each control variable. |
Thanks, reviewers! We appreciate you bringing to our attention that Figure 2 does not display sufficient information. So we have remove it according to your request. Many thanks to your kind advice! Table 2 can serve as a replacement for Figure 3, and we will proceed to remove Figure 3 accordingly. The measurement methods for some variables are shown in Table 2. We have highlighted it accordingly. |
||||||||
|
8. In table 3, it is quite standard to report further indicators such as standard error, Skewness and Kurtosis. |
Thank you for considering our advice! We have incorporated skewness and kurtosis into Table 3 to provide readers with a more comprehensive understanding of our research. |
||||||||
|
9-10. The econometric model (2) is very well presented. Nevertheless, I am afraid that standard tests should have been performed before reaching the final estimation regression equation. These tests include linearity Vs non-linearity (RESET test), Fixed Vs Random Effects (using Hausman test), Heteroskedasticity tests (using Breusch-Pagan / Cook-Weisberg test). Although, tables 7 and 8 show that the authors have reached the estimations assuming fixed effects, there must be evidence that fixed rather than random effect is the appropriate estimation that fits the data. |
Thank you for your advice! We have reconsidered the three issues you mentioned. Specifically, our response to each of the three issues is as follows. (1) linearity Vs non-linearity We conducted the RESET test, and the results are as follows: the F-value is 0.51, and the corresponding p-value is 0.6765. Based on these results, we cannot reject the original hypothesis, indicating that the inclusion of nonlinear higher-order terms in the predicted values does not provide additional information. Therefore, to some extent, using the linear model is appropriate. Table 1. The results of RESET test
(2) Fixed Vs Random Effects We conducted the Hausman test and obtained the following results: the chi-square value is 215.77, with a corresponding p-value of 0.000. Based on these results, we reject the null hypothesis. Consequently, we proceed to estimate the model using fixed effects. Table 2. The results of Hausman test
(3) Heteroskedasticity tests Heteroscedasticity is an important issue that deserves consideration, and we have taken note of it. Referring to Bertrand et al. (2004), we have employed clustering robust standard errors in our regression analysis to mitigate the influence of heteroscedasticity. This approach helps to account for potential heteroscedasticity and provides more accurate and reliable standard errors for the estimated coefficients. By using this method, we aim to address the issue of heteroscedasticity to a considerable extent. Reference [1] Bertrand, M., Duflo, E., & Mullainathan, S. (2004). How Much Should We Trust Differences-in-Differences Estimates? The Quarterly Journal of Economics, 119(1), 249-275. |
||||||||
|
11. The tables do not show the significance (p-value) of each model. For example, in table 7, model 5 is insignificant, thus no further discussion is required. The same is true in table 8, model 4. And in Table 10 model 2. In these cases, the authors must discuss (preferably with proper references) the implications of those insignificant models. |
Thanks, reviewers! We conducted a placebo test in table 7 column (5). We assume that the pilot year is advanced by three years and an experimental group is generated. That is to say, this is a randomly fabricated experimental group. If this is significant, it indicates that our results are not robust. And the results indicate that the coefficient is not significant here which meets our expectations. We conducted heterogeneity analysis in table 8. We have discovered some interesting facts that the significance of coefficients varies for different groups. Specifically, at low levels of intellectual property judicial protection and digitization, the coefficient is not significant, indicating that IPPF has no significant impact on enterprise innovation. We will provide a more detailed explanation of this in the main text. Thanks for pointing out again! |

Reviewer 2 Report
The comments of Reviewer 1 are included in the attached file.

Author Response
Responses to the reviewers
Intellectual Property Pledge Financing and Enterprise Innovation: Based on the Perspective of Signal Incentive
Reviewer 2:
|
Comments |
Responses |
|
1. Row 86: For the sake of modesty in this kind of work, I would suggest replacing the sentence with "This study offers several contributions". |
Thanks for your advice! We have made modifications according to your suggestion. Thank you again for your suggestion! |
|
2.The first paragraph of Section 2.1 should not be italicised |
I’m sorry that we have made an error here. Thanks for pointing out! We have corrected it. |
|
3. Row 142: Avoid ending the sentence and start the following sentence with the numbering of the authors "[11].[12]". I would suggest starting the following sentence with Fujita et al. [12] examined… |
Thanks, reviewers! We have made the modifications according to your request and have also made similar changes to the rest of the text. Thanks for your kind advice! |
|
4. Row 172: I would suggest “The Impact of IPPF on external financial resources”. |
Thanks, reviewers! We have made modifications according to your suggestion. |
|
5. Row 267: The authors could clarify the meaning of "ST, *ST, and SST enterprises”. |
Thanks, reviewers! ST represents the enterprise's two consecutive years of operating losses, with special treatment. * ST represents a enterprise that has suffered losses for three consecutive years and is subject to delisting warning. SST stated that the enterprise has been operating at a loss for two consecutive years, with special treatment, and has not yet completed the share reform. We have made the necessary additions in the main text as requested. Thanks for your valuable advice again. |
|
6. The authors could homogenise the use of symbology of the variables in the text, taking into account that some of them are presented in Table 2. For example, in line 344, the variables Long and Longratio are symbolised, but the variables Risk and Investor Sentiment were not symbolised in the text |
Thanks, reviewers! I’m sorry that this representation may cause confusion. With your suggestion, we have made modifications and the entire text is uniformly represented by representative variables. |
|
7. Table 2: It is up to the authors to specify the units of measurement of the variables. For example, is the GDP variable used in absolute terms or its logarithm? The question arises because of the discrepancy between its possible absolute values and other variables in the model, such as ROA |
Thank you for your suggestion! We have logarithmically transformed all absolute variables and applied winsorization to them (as mentioned in section 3.1). It is necessary to provide a detailed explanation of these procedures in Table 2. We appreciate your valuable advice! |
|
8. Table 6: Authors should add "* for 10% significance" in the legend. |
We have added it. Thanks! |
|
9. Rows 631-634: I think there is confusion between the interpretations of Table 9 and Table 10. |
Thanks, reviewers! We have reinterpreted the content of the table. |
|
10.Row 638: the "Discussion and Conclusion" should be chapter 5. |
Thanks for pointing out! I’m sorry that we have made an error here. We have corrected it. |
|
11. I would suggest a brief justification for the choice of the study period (2007 to 2017) |
Thank you for your valuable request! We chose this specific time interval because, starting from 2017, certain regions in China initiated small-scale IPPF pilot projects. However, these projects have not yet reached a significant scale, and thus we were unable to obtain a comprehensive list of them. To ensure accuracy and avoid potential errors stemming from these small-scale pilot projects conducted after 2017, we have limited our time interval to the year 2017. We have mentioned this in Table 1 and highlighted it in the text. |
|
12.The formulation of each research hypothesis could have been supported by more international empirical evidence on each topic, which would have allowed a confrontation with the results obtained. |
Many thanks to your valuable advice! We have made modifications in the original text based on your suggestion. You can review them. Thank you very much! |
|
13. Line 356: I would suggest improving the explanation of the relationship between Equation (1), where neither the regression results nor their interpretation are presented, and investor sentiment. |
Thank you, reviewers! This equation is used to calculate the value of investore sentiment (IS). And Referring to Rhodes-Kropf et al. (2004), we standardized the residuals to obtain investor sentiment indicators. We did not provide regression results and we only use its calculation results for the next step of analysis. We have included the recommended revisions in the main text to provide interpretation. Thank for your question! We appreciate it!
Reference [1] Rhodes-Kropf, M.; Viswanathan, S.; Robinson, D.T. Valuation Waves and Merger Activity: The Empirical Evidence. 2004. |
|
14. Hypotheses H2 and H3 are defined in terms of "IPPF promotes enterprise innovation through/by", but the underlying equations (3), (4) and (5) do not include the enterprise innovation (RD) variable, either independently or relatedly. In these equations the relationship is established between IPPF and Loans and IS, in addition to the control variables. I would like the authors to reflect on the definition of these hypotheses (and for the same reason on H4 and H5). However, I think the issue could be circumvented if the authors justified the acceptance of the hypotheses by highlighting the previous results (Tables 5, 6 and 7) on the relevance of the IPPF variable in estimating RD |
Thanks, reviewers! We are very grateful for your valuable suggestion! We attempt to explain using a Mediating Effect Model in mechanism analysis section. That is, IPPF will affect RD by influencing mediating variables (i.e. myopia, risk, long, longratio and IS). We believe that the relationship between mediating variables and RD does not require further testing, while the relationship between IPPF and mediating variables needs to be tested (For example, will IPPF affect executive behavior, which is unclear before testing). Therefore, we did not test RD in the research hypothesis section and mechanism analysis section. We should follow your suggestion and add the logical relationship between mediating variables and RD. We have made revisions and supplements, and thank you again for your kind suggestion! |

Round 2
Reviewer 2 Report
I was pleased with the authors' responses and the improvement of the manuscript. I would like to wish the authors success in their research work.